# Algorithm-Agnostic Explainability for Unsupervised Clustering

## Abstract

Supervised machine learning explainability has developed rapidly in recent years. However, clustering explainability has lagged behind. Here, we demonstrate the first adaptation of model-agnostic explainability methods to explain unsupervised clustering. We present two novel "algorithm-agnostic" explainability methods – global permutation percent change (G2PC) and local perturbation percent change (L2PC) – that identify feature importance globally to a clustering algorithm and locally to the clustering of individual samples. The methods are (1) easy to implement and (2) broadly applicable across clustering algorithms, which could make them highly impactful. We demonstrate the utility of the methods for explaining five popular clustering methods on low-dimensional synthetic datasets and on high-dimensional functional network connectivity data extracted from a resting-state functional magnetic resonance imaging dataset of 151 individuals with schizophrenia and 160 controls. Our results are consistent with existing literature while also shedding new light on how changes in brain connectivity may lead to schizophrenia symptoms. We further compare the explanations from our methods to an interpretable classifier and find them to be highly similar. Our proposed methods robustly explain multiple clustering algorithms and could facilitate new insights into many applications. We hope this study will greatly accelerate the development of the field of clustering explainability.

## 1   Introduction

In recent years, research into explainability for supervised learning methods has greatly accelerated. However, relatively little research into methods for explaining unsupervised clustering has occurred, and more approaches need to be developed. Adapting and expanding model-agnostic explainability methods from the domain of supervised learning to explain clustering algorithms is a previously unexplored avenue for accelerating the development of the field of clustering explainability. In this study, we demonstrate-for the first time-the viability of this avenue, and introduce two novel clustering explainability methods that are expansions of existing methods for supervised learning. Importantly, these methods are easy to implement and applicable to many clustering algorithms, making them ideal for widespread use.

Explainability methods for supervised learning fall into two categories - model-specific and model-agnostic. It should be noted that these explainability methods are distinct from inherently interpretable machine learning methods like logistic regression, which was introduced by Cox (1958), or decision trees. Model-specific methods are only applicable to a specific class of models. For example, layer-wise relevance propagation (LRP) is specific to differentiable classifiers, as explained in Bach et al. (2015), and impurity-based feature importance is specific to decision tree-based classifiers (i.e., Louppe (2014)). In contrast, model-agnostic methods are applicable to a variety of supervised models. Examples of model-agnostic methods include LIME by Ribeiro et al. (2016), SHAP by Lundberg & Lee (2017), permutation feature importance by Fisher et al. (2018), PD plots by Friedman (2001), and ICE plots by Goldstein et al. (2015). Permutation feature importance is unique among these methods for two reasons. (1) It is easy to implement, and (2) it can be used with classifiers that only provide hard classifications. Most other model-agnostic methods are somewhat difficult to implement and require that a classifier output a probability of belonging to a class. We explain the relevance of these two key differences later in this paper.

It is also worth noting that permutation feature importance is a global method and not a local method. Global methods provide insight into the features that are generally prioritized by a classifier (e.g., impurity, PD plots), and local methods provide insight into the features that are important for the classification of an individual sample (e.g., LRP, LIME, SHAP, and ICE plots). Perturbation, another explainability method, is easy to implement like permutation feature importance and offers local insight. However, like most model-agnostic explainability approaches, it has previously only been used with classifiers that provide soft labeling. Examples of studies using perturbation include works by Ellis et al. (2021) and Fong & Vedaldi (2017).

Methods analogous to model-specific explainability and interpretable machine learning have been developed for clustering. While many interpretable clustering methods use decision trees, such as Basak & Krishna-puram (2005); Bertsimas et al. (2018; 2020); Fraiman et al. (2013); Loyola-González et al. (2020), other more distinct approaches have also been developed by Bhatia et al. (2019) and Plant & Böhm (2011). Some methods have properties of both model-specific explainability and interpretable machine learning like an explainable K-means clustering approach by Frost et al. (2020), and some methods, like the unsupervised K-means feature selection approach by Boutsidis et al. (2009), are feature selection methods that are analogous to model-specific explainability. However, to the best knowledge of the authors, no existing explainability approaches have been developed that are broadly applicable to many different clustering algorithms, and most existing clustering methods still remain unexplainable. This is problematic because many traditional clustering methods like density-based clustering in Sander (2010) have unique characteristics. These unique characteristics make them ideal for specific use-cases, for which existing interpretable cluster methods are suboptimal. Model-agnostic explainability methods offer a solution to this problem, and their potential still remains untapped within the space of explainable clustering. If slightly adapted, they could be directly transferred from the domain of supervised learning to unsupervised clustering and could greatly accelerate the development of the field of explainable clustering. We call this new class of unsupervised explainability methods, "algorithm-agnostic".

There are multiple possible taxonomies of clustering methods, and two common taxonomies are described in Fahad et al. (2014); Rai (2010). For the purposes of this study however, we consider how model-agnostic explainability methods can be applied to five types of clustering methods: (1) partition-based, (2) density-based, (3) model-based, (4) hierarchical, and (5) fuzzy methods. These methods are respectively described in Jin & Jan (2010); Sander (2010); Banerjee & Shan (2010); Johnson (1967), and Ruspini et al. (2019). Although clustering has been applied across many domains in works like Behera & Panigrahi (2015); Mustaniroh et al. (2018), and Thomas et al. (2018), in this work, we validate our approaches within the field of computational neuroscience for resting-state functional magnetic resonance imaging (rs-fMRI) functional network connectivity (FNC) analysis. Clustering approaches have been applied to FNC data to gain insight into a variety of brain disorders and mechanisms in works by Sendi et al. (2020; 2021a;b) and Zendehrouh et al. (2020). However, the high dimensionality of the data makes understanding clusters extremely challenging, and as a result, most studies only examine a small number of domains and train a supervised machine learning classifier after clustering to gain insight into the identified clusters.

Here, we demonstrate how model-agnostic explainability methods can be generalized to the domain of explainable clustering by adapting permutation feature importance, a method described in Fisher et al. (2018)). Importantly, we adapt permutation feature importance for the two distinguishing characteristics that we previously described. An algorithm-agnostic version of permutation feature importance could easily be implemented by researchers with varying levels of data science expertise and thus be widely adopted across many domains. Moreover, it could provide explanations for both hard and soft clustering methods, unlike most model-agnostic methods that could only explain clustering methods with soft labeling. As such, permutation feature importance should be more widely generalizable to clustering methods than other model-agnostic explainability approaches. We also adapt perturbation, a local model-agnostic explainability method, to provide explanations for clustering algorithms. This adaptation is particularly innovative because perturbation can typically only be applied to methods with soft labeling, and we describe a novel approach that enables it to be applied to methods that use hard labeling. Based upon these adaptations, we present two novel methods for algorithm-agnostic clustering explainability: Global Permutation Percent Change (G2PC) feature importance and Local Perturbation Percent Change (L2PC) feature importance. G2PC provides "global" insight into the features that distinguish clusters, and L2PC provides "local" insight

into what makes individual samples belong to a particular cluster. We demonstrate the utility of these approaches for the previously mentioned five categories of clustering methods on low-dimensional synthetic data and further demonstrate how they can provide insight into high-dimensional rs-fMRI FNC data from the Functional Imaging Biomedical Informatics Research Network (FBIRN) dataset which includes 151 subjects with schizophrenia and 160 controls. Further details on the dataset are described in van Erp et al. (2015). We compare their explanations to those of an interpretable machine learning classifier to better understand the reliability of each method.

## 2 Methods

Here we describe (1) our proposed clustering explainability methods and (2) the experiments with which we examine their utility.

### 2.1 Explainability methods for clustering

We first discuss permutation feature importance, a model agnostic explainability method, from which G2PC was derived. We then discuss G2PC and L2PC.

#### 2.1.1 Permutation feature importance

Permutation feature importance, a form of feature perturbation, is a well-known model-agnostic explainability method that is typically applied within the context of supervised machine learning algorithms. It was originally developed for random forests by Breiman (2001) and was later expanded to be model-agnostic by Fisher et al. (2018). It involves a straightforward procedure for estimating feature importance that is visualized in algorithm 1.

---

**Algorithm 1** Permutation Feature Importance

---

1: **function** PERMUTE_FEATURE($X_2, j, features$)
2:     $p \leftarrow$ PERMUTE($0$ **to** $N - 1$)                                   ▷ p - permuted indices, N - number of samples
3:     $X_2[:, features = j] \leftarrow X_2[p, features = j]$
4:     **return** $X_2$
5: **end function**
6:
1: **procedure** PERMUTATION FEATURE IMPORTANCE($model, X, Y$)                         ▷ X - data, Y - labels
2:     $Y_1 \leftarrow$ PREDICT(X)
3:     $performance_1 \leftarrow$ PERFORMANCE($Y, Y_1$)
4:     **for** $j$ in J features **do**                                              ▷ J - number of features
5:         **for** $k$ in K repeats **do**                                          ▷ K - number of repeats
6:             $X_2 \leftarrow$ COPY(X)
7:             $X_2 \leftarrow$ PERMUTE_FEATURE($X_2, j, features$)
8:             $Y_2 \leftarrow$ PREDICT($X_2, model$)
9:             $performance_2 \leftarrow$ PERFORMANCE($Y, Y_2$)
10:            $Importance[j, k] \leftarrow (performance_2 - performance_1) / performance_1$
11:        **end for**
12:    **end for**
13:    **return** $Importance$                                                     ▷ Results
14: **end procedure**

---

#### 2.1.2 Global permutation percent change (G2PC) feature importance

The permutation feature importance approach can be generalized to provide an estimate of feature importance for clustering algorithms. G2PC feature importance is highly similar to the standard permutation

feature importance applied to supervised machine learning models. However, there are several key distinctions. Rather than calculating the ratio of the change in performance before and after permutation, G2PC calculates the percentage of all $N$ samples that change from their original pre-permutation clusters to a different post-permutation cluster. The percentage of samples that switch clusters following the permutation of a particular feature reflects the importance of that feature to the clustering, and this metric is one of the key novelties of our adaptation. We also added a grouping component to G2PC that allows related features to be permuted simultaneously. Figure 1 and Algorithm 2 portray the method in greater detail. The *groups* parameter is an array that contains an identifying group number for each feature. The *mdl* parameter is an object that contains the information necessary to assign a new sample to the preexisting clusters. Parameters $X$ and $C$ are the original data and cluster assignments, respectively. *Pct_Chg* stores the calculated percent change importance values.

---

**Algorithm 2** G2PC: Global Permutation Percent Change Feature Importance

---

 1: **function** PERMUTE_GROUP($X_2, j, groups$)
 2:     $p \leftarrow$ PERMUTE($0$ **to** $N - 1$)
 3:     $X_2[:, groups = j] \leftarrow X_2[p, groups = j]$
 4:     **return** $X_2$
 5: **end function**

 6:
 1: **procedure** G2PC($model, X, C, groups$)        ▷ $X$ - data, $C$ - cluster assignments, $groups$ - group each feature belongs to
 2:     **for** $j$ in J groups **do**
 3:         **for** $k$ in K repeats **do**
 4:             $X_2 \leftarrow$ COPY(X)                            ▷ $X_2$ - copy of samples to permute
 5:             $X_2 \leftarrow$ PERMUTE_GROUP($X_2, j, groups$)
 6:             $C_2 \leftarrow$ RECLUSTER($X_2, model$)                        ▷ See Section 2.1.4
 7:             $Pct\_Chg[j, k] \leftarrow$ SUM(C $\neq C_2$) / N        ▷ $C_2$ - perturbed sample cluster assignments
 8:         **end for**
 9:     **end for**
10:     **return** $Pct\_Chg$                                    ▷ Results
11: **end procedure**

---

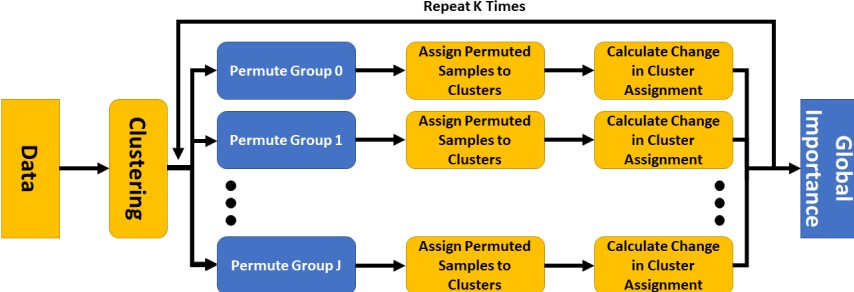

Figure 1: Diagram of G2PC. For G2PC, the data is first input into a clustering algorithm. After clustering, each of $J$ feature groups is permuted across samples, and the resulting perturbed samples are reassigned to one of the previously identified clusters. Then the percent of samples that change cluster assignment is calculated to give insight into the relative impact of permuting that group of features, and the process is repeated $K$ times. Rectangles with rounded corners reflect operations, and rectangles without rounded corners reflect inputs or outputs. Gold rectangles refer to operations or inputs that are identical across both G2PC and L2PC, and blue boxes refer to operations or outputs that are distinct across methods.

### 2.1.3   Local perturbation percent change (L2PC) feature importance

L2PC feature importance extends perturbation to provide explainability for clustering algorithms. However, it is related to G2PC. Rather than permuting across all samples and measuring the overall clustering percent change, L2PC swaps (i.e., perturbs) each of $J$ features of a sample with values randomly selected from the same feature of other samples in the dataset $M$ times and calculates the percentage of times that the sample changes clusters following the perturbations. It performs this operation for a pre-defined number of repeats ($K$) per sample. Importantly, using the percent of perturbations that cause a change in cluster assignment, rather than measuring the change in classification probability associated with a perturbation, enables us to extend perturbation to explain methods that employ hard labeling or cluster assignments. The perturbation percent change values obtained from each repeat could be used to obtain the statistical significance of each feature by comparing the values to a null hypothesis of zero perturbation percent change. As the percent change increases, the importance of a feature for the clustering of the individual sample increases. While L2PC is principally a method for obtaining a measure of feature importance for each sample individually, L2PC can be applied to each sample in a data set to obtain a global measure of feature importance like other local methods. The mean of the resulting distribution of perturbation percent change values across samples can provide a global measure of feature importance. The application of L2PC as a global measure of feature importance is much more computationally intensive than G2PC. However, the computational complexity is not problematic for data sets with a small number of samples. Using L2PC as a global metric in high dimensions could be made more tenable by using a smaller number of repeats or perturbations per repeat, but that would reduce the reliability of the resulting importance estimates. Additionally, the approach can easily be parallelized to greatly decrease its execution time. We also added a grouping component to L2PC that allows related features to be perturbed simultaneously. Figure 2 and Algorithm 3 portray the method in greater detail. The variables used in Algorithm 3 are the same as those used in Algorithm 2.

---

**Algorithm 3** L2PC: Local Perturbation Percent Change Feature Importance

---

1: **function** PERMUTE_GROUP($X_2, X, M, j, groups$)
2:     $p \leftarrow$ PERMUTE($0$ **to** $N - 1$)                                    ▷ generate permuted indices
3:     $X_{perturb}[:, groups = j] \leftarrow X[p < M - 1, groups = j]$     ▷ get samples at permuted indices below $M$
4:     **return** $X_{perturb}$
5: **end function**
1: **procedure** L2PC($model, X, C, groups$)
2:     **for** $j$ in J groups **do**
3:         **for** $n$ in N samples **do**
4:             **for** $k$ in K repeats **do**
5:                 $X_{perturb} \leftarrow$ COPY(X$[n, :]$)                                    ▷ M x F
6:                 $X_{perturb} \leftarrow$ PERMUTE_GROUP($X_{perturb}, X, M, j, groups$)
7:                 $C_2 \leftarrow$ RECLUSTER($X_{perturb}, model$)                         ▷ See Section 2.1.4
8:                 $Pct\_Chg[n, j, k] \leftarrow$ SUM(C$[n] \neq C_2$) / M
9:             **end for**
10:         **end for**
11:     **end for**
12:     **return** $Pct\_Chg$                                                            ▷ Results
13: **end procedure**

---

### 2.1.4   Additional notes on G2PC and L2PC

It should be noted that one component of both G2PC and L2PC is the assignment of new samples to existing clusters. Some might argue that the assignment of new samples to existing clusters violates the intended purpose of clustering methods. Regardless, it is still possible to effectively assign new samples to existing clusters without the use of a supervised classifier. For k-means, samples can be assigned to the cluster with the nearest center. For Gaussian mixture models (GMMs), the locations of a new sample within the existing cluster probability density functions can be obtained, and the sample can be assigned to the cluster with the

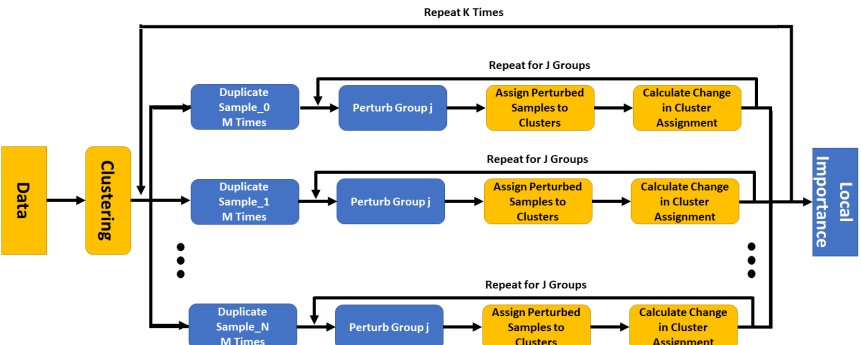

Figure 2: Diagram of L2PC. The input data is assigned to clusters. Then each sample is duplicated $M$ times, and the feature group j of each duplicate sample is perturbed by values randomly selected from the same feature group of other samples in the dataset. The perturbed duplicate samples are then reassigned to the previously identified clusters, and the percent of times that the perturbed sample switches cluster assignments is calculated. The process is repeated for each of $J$ feature groups and $K$ times. Rectangles with rounded corners reflect operations, and rectangles without rounded corners reflect inputs or outputs. Gold rectangles refer to operations or inputs that are identical across both G2PC and L2PC, and blue boxes refer to operations or outputs that are distinct across methods.

highest probability. For fuzzy c-means, new samples can be assigned to existing clusters by retraining a new clustering model with fixed cluster centers and parameters identical to those of the original clustering. K-means, Gaussian mixture models, and fuzzy c-means clustering have functions for predicting the assignment of new samples in scikit-learn by Pedregosa et al. (2011) and scikit-fuzzy[1]. For DBScan, new samples can be assigned to the cluster of a core sample that is within a pre-defined $\varepsilon$ distance, and new samples can be assigned to clusters derived from agglomerative clustering by placing them in the cluster of the nearest sample.

## 2.2 Description of experiments

We evaluate the performance of the two explainability methods through a series of 3 experiments. The first two experiments involve the use of low-dimensional, synthetic, ground-truth data, and the last experiment involves the application of the methods to functional network connectivity values extracted from the FBIRN rs-fMRI data set that is described in van Erp et al. (2015). We apply five popular clustering methods to each dataset and apply both novel explainability methods to evaluate the relative importance of each feature to the clustering.

### 2.2.1 Synthetic datasets

We generated two synthetic datasets with different numbers of clusters and random distributions.

Table 1: Synthetic dataset distributions

|         | Synthetic dataset 1 | | Synthetic dataset 2 | | | |
|---------|-------------|-------------|-------------|-------------|-------------|-------------|
| Feature | Class 1     | Class 2     | Class 1     | Class 2     | Class 3     | Class 4     |
| 1       | $11 \pm 1$  | $3 \pm 1$   | $3 \pm 0.5$ | $11 \pm 0.5$| $19 \pm 0.5$| $27 \pm 0.5$|
| 2       | $9 \pm 1$   | $3 \pm 1$   | $3 \pm 0.5$ | $9 \pm 0.5$ | $15 \pm 0.5$| $21 \pm 0.5$|
| 3       | $7 \pm 1$   | $3 \pm 1$   | $3 \pm 0.5$ | $7 \pm 0.5$ | $11 \pm 0.5$| $15 \pm 0.5$|
| 4       | $5 \pm 1$   | $3 \pm 1$   | $3 \pm 2$   | $5 \pm 2$   | $7 \pm 2$   | $9 \pm 2$   |
| 5       | $3 \pm 1$   | $3 \pm 1$   | $3 \pm 2$   | $4 \pm 2$   | $5 \pm 2$   | $6 \pm 2$   |

---

[1]https://github.com/scikit-fuzzy/scikit-fuzzy

**Synthetic dataset 1** Synthetic dataset 1 consists of 5 features with two 50-sample clusters. The features – numbered 1 through 5 - each consist of random variables generated from two separate normal distributions that formed two clusters. As the feature number increases (i.e., from feature 1 to feature 5), the difference between the means ($\mu$) of the two random variable normal distributions within each feature decreases, meaning that the overall expected importance of the features decrease. The standard deviation ($\sigma$) of the random variables is consistent across both clusters and all 5 features. Table 1 shows the $\mu$ and $\sigma$ of the random variables. To test G2PC, we generated 100 sets of simulated data, and to test L2PC, we generated 1 set of simulated data. Figure 3 depicts synthetic dataset 1 following dimensionality reduction with t-SNE.

**Synthetic dataset 2** Synthetic dataset 2 consists of five features with four 50-sample clusters. It provides an opportunity to examine the behavior of the explainability methods in a context involving more than two clusters. The features–numbered 1 through 5-each consists of random variables generated from four separate normal distributions that formed four clusters. As the feature number increase, the difference between the $\mu$ of each of the random variables within the features decreases. Additionally, the variance of the first three features is below unit variance, and the variance of the last two features is above unit variance. As such, we expect the feature importance methods to assign decreasing importance from Feature 1 to Feature 5 and to assign significantly less importance to Features 4 and 5 than Features 1 through 3. Table 1 shows the mean and SD of the random variable distributions of each feature. To test G2PC, we generated 100 sets of simulated data, and to test L2PC, we generated 1 set of simulated data. Figure 3 depicts synthetic dataset 2 following dimensionality reduction with t-SNE.

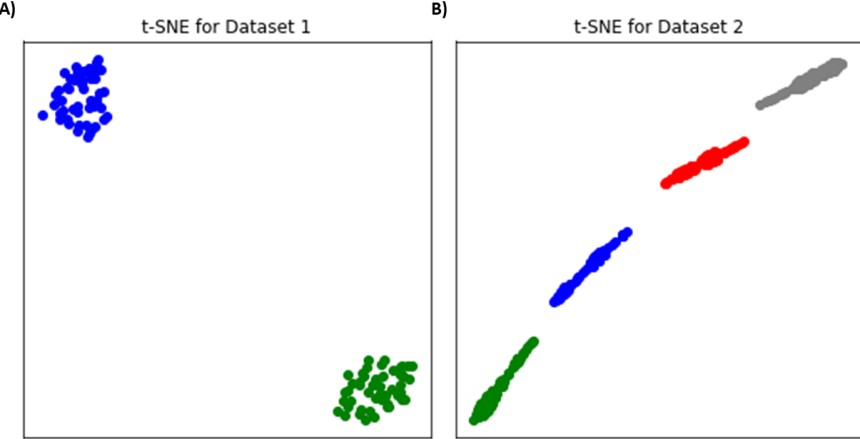

Figure 3: Display of Synthetic Data with t-SNE Dimensionality Reduction. Panel A shows synthetic dataset 1 following dimensionality reduction, and Panel B shows synthetic dataset 2 following dimensionality reduction. Dimensionality reduction is performed using t-SNE, which was introduced by van der Maaten & Hinton (2008), with 3,000 iterations and a perplexity of 35. Each cluster is denoted with a different color.

### 2.2.2 FBIRN rs-fMRI dataset

We use rs-fMRI and clinical data from the FBIRN dataset van Erp et al. (2015). The dataset can be made available upon a reasonable request emailed to the corresponding author and contingent upon IRB approval. The dataset contains 151 schizophrenia (SZ) subjects and 160 healthy controls (HC). Details on the data collection and preprocessing are found in Appendix A. After performing initial preprocessing, we perform group independent component analysis (ICA) and extract 53 independent components (ICs) with Neuromark by Du et al. (2020). We assign the ICs to seven domains, including subcortical network (SCN), auditory network (ADN), sensorimotor network (SMN), visual network (VSN), cognitive control network (CCN), default-mode network (DMN), and cerebellar network (CBN). A full list of all components can be found in Du et al. (2020) and Sendi et al. (2021b). After extracting the ICs, we use Pearson correlation between each IC time-series pair to estimate each participant's static functional network connectivity (FNC). This results in 1378 whole-brain connectivity values (i.e. features) for each individual.

### 2.2.3 Clustering methods

We utilize algorithms from 5 different categories of clustering methods: (1) k-means clustering, a partitioning method, (2) DBScan, a density-based method, (3) GMM, a model-based method, (4) agglomerative clustering (AGC), a hierarchical clustering method, and (5) fuzzy c-means clustering, a fuzzy clustering method. The GMM, AGC, k-means, and DBScan are implemented in scikit-learn, which was developed by Pedregosa et al. (2011), and fuzzy c-means is implemented in scikit-fuzzy[2].

### 2.2.4 Experiment parameters for clustering and explainability methods

For the synthetic datasets, we provide the ground-truth number of clusters to the k-means, GMM, AGC, and fuzzy c-means algorithms. For the rs-fMRI FNC analysis, we optimize the number of clusters for each algorithm via the silhouette method, which was introduced in Kaufman & Rousseeuw (1990). For each DBScan analysis, the $\varepsilon$ distance parameter is optimized using the silhouette method, and the minimum number of points parameter equals 4. For fuzzy c-means in all of the analyses, m = 2, error = 0.005, and maximum number of iterations = 1000. When calculating the percent change in clustering, the cluster of each sample is assigned to the class with the highest predicted likelihood. Both synthetic datasets are z-scored on a feature-wise basis.

The number of repeats ($K$) parameter is set to 100 for the G2PC and L2PC analyses on all datasets, and the number of perturbations per repeat (M) is set to 30 in L2PC for all datasets. We perform G2PC and L2PC on the FBIRN data, both with grouping based upon the established FNC domains and without grouping. To compare our G2PC and L2PC results for the FBIRN dataset to an existing interpretable machine learning method, we train a logistic regression classifier with elastic net regularization (LR-ENR) with the k-means and fuzzy c-means cluster assignments as labels. We train the LR-ENR models with 10 outer and 10 inner folds. In each fold, 64%, 16%, and 20% of the samples are randomly assigned to training, validation and, test sets. After training and testing the classifiers, we output their coefficient values and multiply them by each test sample in their respective fold. We then calculate the the absolute value of the resulting values. This generates a measure of the effect that each feature had upon the model. We then compute the mean effect within each fold and the mean of the mean effect across all features associated with each FNC domain. This provides an estimate of the effect of each domain on the classifier. Python code for all experiments is on GitHub[3]. Note that Figures 5 and 6 are generated separately in MATLAB R2020b (MathWorks, Inc) with the results from the Python scripts.

## 3 Results

Here we detail the results for both the synthetic data and rs-fMRI FNC experiments.

### 3.1 Synthetic datasets

As shown in Figure 4, all clustering methods successfully identify the underlying clusters. Importance generally decreases from features 1 to 5 for synthetic dataset 1 in alignment with expectations for all clustering and both explainability methods. It should be noted, however, that the sensitivity of each of the clustering methods to perturbation varies on a feature-to-feature basis and on an overall percent change basis. For G2PC, features 4 and 5, which are supposed to be unimportant, are generally categorized as having little to no effect upon the clustering. Additionally, importance generally decreases from feature 1 to feature 2 and feature 3. However, GMMs with G2PC seem to mainly emphasize one feature (i.e. feature 1) in their clustering, contrary to the other methods that place more importance upon features 2 and 3. Also, some methods seem much more sensitive to perturbation than others. The mean percent change for feature 1 of both the GMM and DBScan is around 30%, while for feature 1 of k-means, AGC, and fuzzy c-means, the median importance is around 2% to 5%. Similar findings occur for L2PC with Synthetic Dataset 1. As can be determined via the black line on Figure 4, the mean feature importance across all samples and

---

[2]https://github.com/scikit-fuzzy/scikit-fuzzy
[3]https://github.com/cae67/G2PC_L2PC

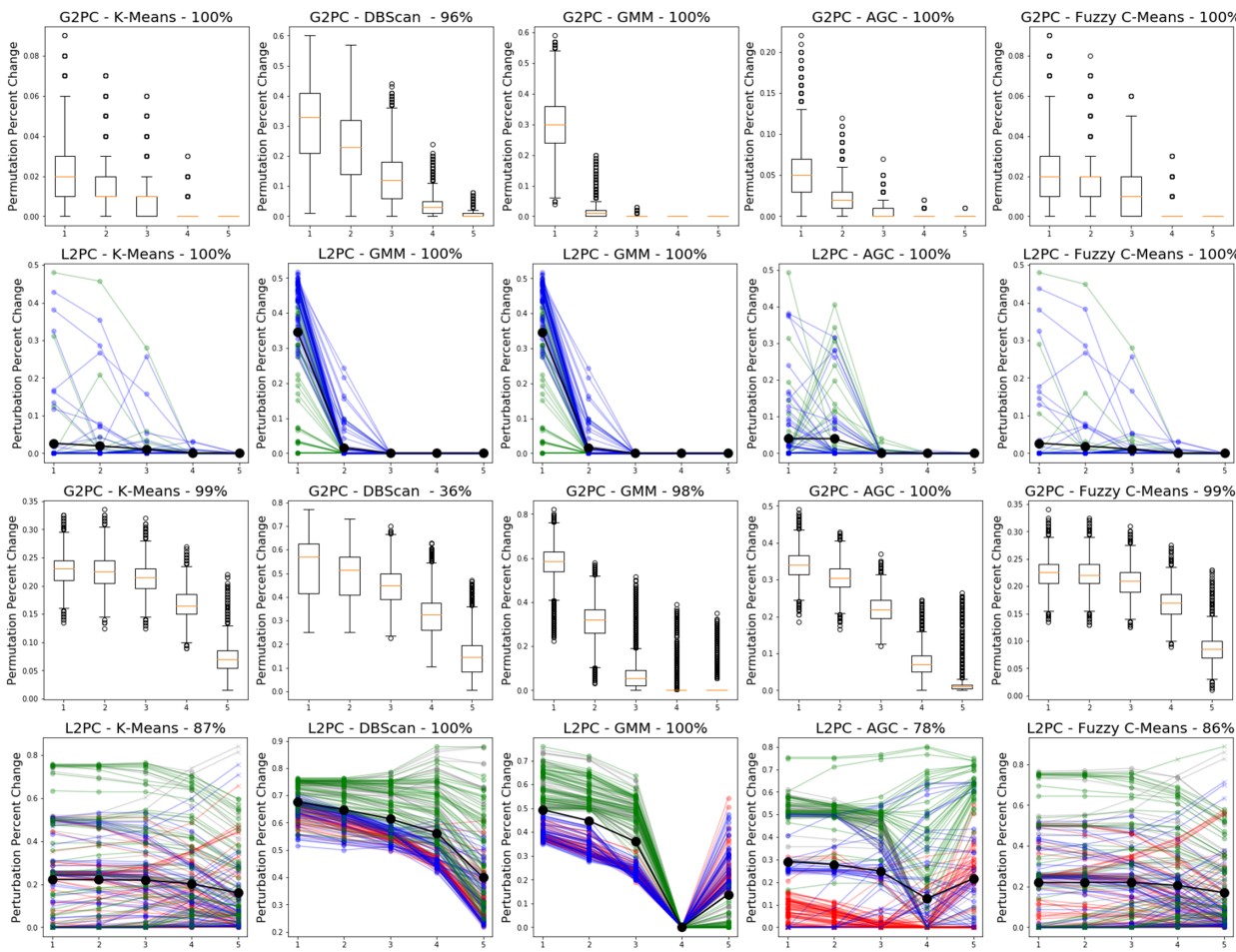

Figure 4: G2PC and L2PC Results for Synthetic Data. The top two rows show the G2PC and L2PC results for Synthetic Dataset 1, and the bottom two rows show the G2PC and L2PC results for Synthetic Dataset 2. Each column shows the results for a different clustering algorithm. Note that as expected, the permutation and perturbation percent change values, in general, decrease from left to right, indicating a relative decrease in feature importance. For the L2PC results, the samples initially belonging to each cluster have different colors. Each line connecting different points reflects an individual sample. Correctly and incorrectly clustered samples are represented by "o" and "x", respectively, and the accuracy of the clustering is indicated by the percentage in the title associated with each subplot. For G2PC, the percentage in the title indicates the mean accuracy across the 100 iterations of datasets.

repeats is very similar to the median values of the G2PC results. For k-means, AGC, and fuzzy c-means, only a few samples are sensitive to perturbation. However, in the GMM and DBScan, a majority of the samples are sensitive to perturbation. It can be seen for some of the clustering methods like fuzzy c-means, AGC, k-means, and the GMM that there are differences in the sensitivity of samples to perturbation on a per-cluster basis.

For Synthetic Dataset 2, most clustering methods have a high level of accuracy when identifying the underlying clusters, as shown in Figure 4. K-means, fuzzy c-means, and AGC do not have 100% accuracy for L2PC. Additionally, DBScan has negligible mean accuracy for G2PC, as across 100 iterations of datasets, it frequently identified less than 4 clusters. The G2PC and mean L2PC results look as expected. The first three features are generally ranked as much more important than the last two features, which accounts for the difference in variance of features 1 to 3 and features 4 to 5. Importance also generally decreases from

features 1 to 3 and 4 to 5, which accounts for the differences in the means of the random variables of each cluster. It is interesting that the GMM and AGC have a sharp increase in mean L2PC importance from feature 4 to 5, while the other three cluster methods have the expected decrease. For AGC, this might be attributed to its inaccurate clustering of some samples. However, for the GMM, it is unclear what may be the cause. It may be attributable to the random initialization of the data, as the GMM L2PC values defer markedly from the G2PC values. The samples in some clusters of the GMM and AGC L2PC results are highly distinct. For the GMM, the red and grey clusters have higher importance for features 1 to 3 and lower importance for features 4 and 5 than the red and blue clusters. For features 1 to 3 of AGC, the red and grey clusters have high importance, the blue cluster have moderate importance, and the green cluster have low importance. It is interesting that the L2PC results show increased variance for features 4 and 5 for most of the clustering methods and that the values do not seem to be cluster dependent. Given that many clustering methods can identify clusters regardless of whether distinct clusters actually exist and that the samples in features 4 and 5 are very dense, it is possible that smaller perturbations of those features may affect some samples more than others.

### 3.2 FBIRN rs-fMRI data

All clustering methods, except for DBScan which only finds noise points, identify 2 clusters as optimal. The identification of two clusters is consistent with the underlying SZ and HC groups. K-means has an accuracy, sensitivity, and specificity of 63.99%, 78.81%, and 52.98%, respectively. Fuzzy c-means has better clusters with an accuracy, sensitivity, and specificity of 68.81%, 74.17%, and 67.55%, respectively.

Permutation and perturbation do not have a widespread effect upon the clustering when each of the 1378 whole-brain correlation values are perturbed individually. However, when we permute or perturb the whole-brain correlation values within each set of inter-domain or intra-domain features simultaneously, we obtain feature importance results for k-means and fuzzy c-means that can be seen in Figure 5. Panels A and B are box plots of the G2PC importance values for each of the 100 repeats, and for the sake of easy visualization, panels C and D reflect the mean perturbation percent change across all repeats and perturbation instances for each sample, where the green samples and the blue samples belong to the SZ dominant cluster and the HC dominant cluster, respectively. Panels E and F show the mean effect of each domain upon the LR-ENR classifiers. Figure 6 shows the values of the mean FNC for the SZ dominant cluster minus the mean FNC for the HC dominant cluster. The domains surrounded by white boxes were those identified by G2PC and L2PC as most important. The figure uses the magma colormap, as described by Biguri (2021).

The feature importance results are highly consistent across both clustering methods and all explainability methods. LR-ENR classifies the samples as their assigned clusters with a $\mu$ area under the receiver operating characteristic curve (AUC) of 99.86 and $\sigma$ of 0.14 for k-means and a $\mu$ AUC of 99.48 and $\sigma$ of 0.63 for fuzzy c-means, indicating that LR-ENR can likely identify the patterns learned by the clustering algorithms. The similarity between the resulting importance estimates for LR-ENR and those for G2PC and L2PC supports the validity of our novel methods. Top domains identified by k-means with G2PC and L2PC include the cross-correlation between the ADN and CCN (i.e., ADN/CCN), the SCN/VSN, the CCN/DMN, and the SMN/DMN. While LR-ENR generally agrees on these top domains, it does not find SMN/DMN to be very important. Top domains identified by fuzzy c-means with G2PC, L2PC, and LR-ENR include the ADN/CCN, the CCN/DMN, the SCN/VSN, ADN, and SMN/VSN. For k-means, relative to the HC dominant group, the SZ dominant group has higher ADN/CCN FNC for around half of the domain, much higher levels of SCN/VSN FNC, a mixture of moderately higher to lower levels of CCN/DMN FNC, and a mixture of higher to lower SMN/DMN connectivity. For fuzzy c-means, relative to the HC dominant group, the SZ dominant group has ADN/CCN FNC values that are higher for around half of the domain, a mixture of higher to lower CCN/DMN FNC values, much higher SCN/VSN FNC values, slightly lower ADN FNC values, and much smaller SMN/VSN FNC values.

Both G2PC and L2PC indicate that a minority of samples are sensitive to perturbation, and L2PC demonstrates that many of the samples that are sensitive are very sensitive. This may indicate that those samples are closer to the boundaries of their clusters or inherently noisier for some reason. For fuzzy c-means, more samples belonging to the SZ dominant cluster rather than the HC dominant cluster seem sensitive to perturbation. Most of the HC samples that are sensitive to perturbation seem to be correctly assigned to the

HC dominant cluster. In contrast, most of the subjects in the SZ dominant cluster that are sensitive to perturbation seem to actually be HC subjects. For k-means, some samples in the HC dominant cluster that are actually SZ subjects seem to be extremely sensitive to perturbation, while other samples belonging to the SZ dominant cluster that are actually HC subjects are sensitive to perturbation at smaller levels.

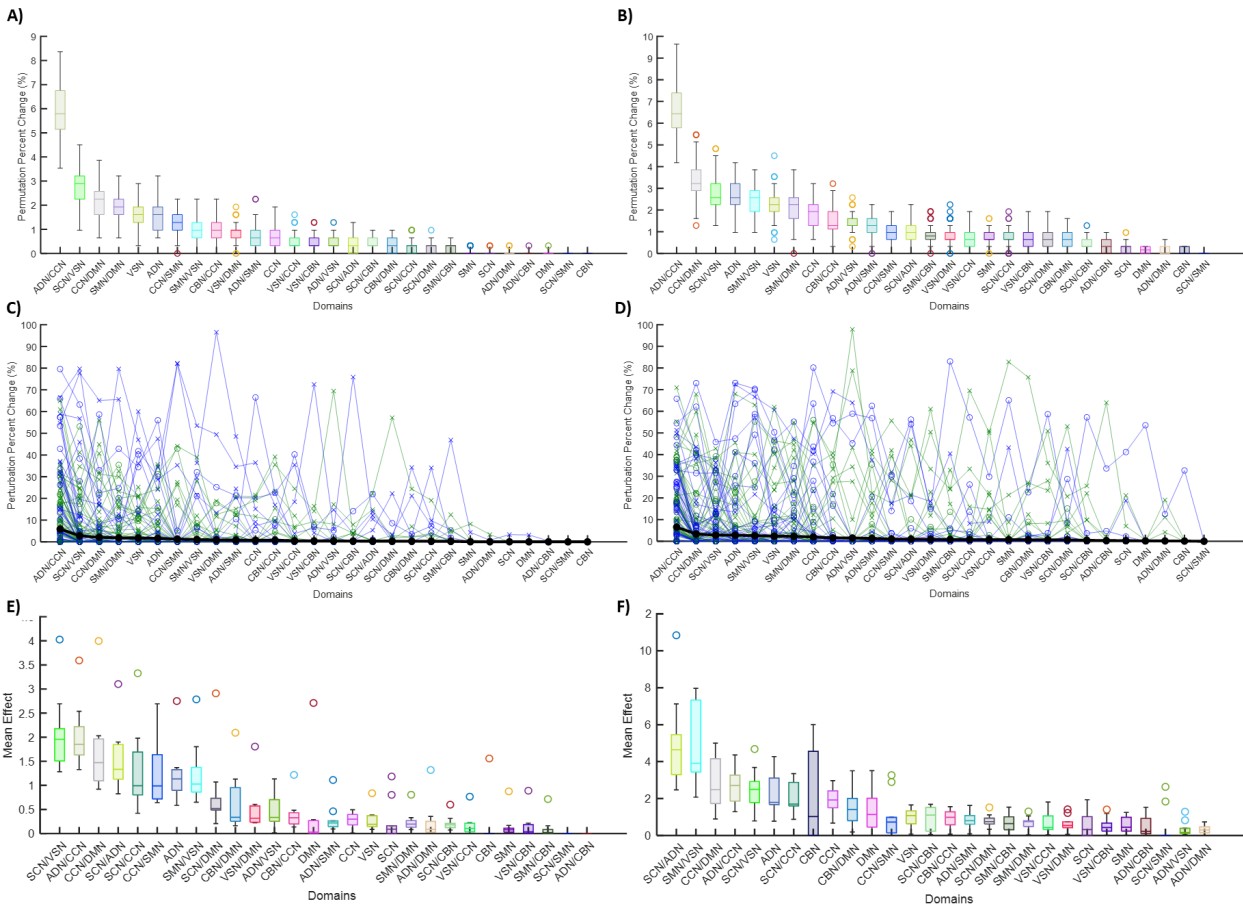

Figure 5: FBIRN G2PC and L2PC Results. Panels A and B show the G2PC results for k-means and fuzzy c-means clustering, respectively, and Panels C and D show the L2PC results for k-means and fuzzy c-means clustering, respectively. Panels E and F show the mean effect results for logistic regression across 10 folds. The x-axis of each panel shows the domains in order of most important to least important based upon their mean value, and the y-axis shows the perturbation or permutation percent change or mean effect. In panels C and D, the samples belonging to the SZ and HC dominant clusters are green and blue, respectively. The values marked with an "o" or an "x" indicate the samples that were correctly or incorrectly clustered, respectively. The black line on Panels C and D reflects the mean L2PC values across all subjects and provides a measure of global feature importance. The lines connecting different points reflect individual samples.

## 4    Discussion

In this study, we propose that model-agnostic methods for supervised machine learning explainability can be adapted for use in unsupervised clustering explainability. We further demonstrate two novel approaches for clustering explainability: G2PC feature importance and L2PC feature importance. G2PC provides a global measure of the relative importance of features to the clustering, and L2PC provides a local measure of which features are important to the clustering of an individual sample. Our adaptation of permutation feature importance and perturbation provides our methods with two unique capabilities. (1) They are easy

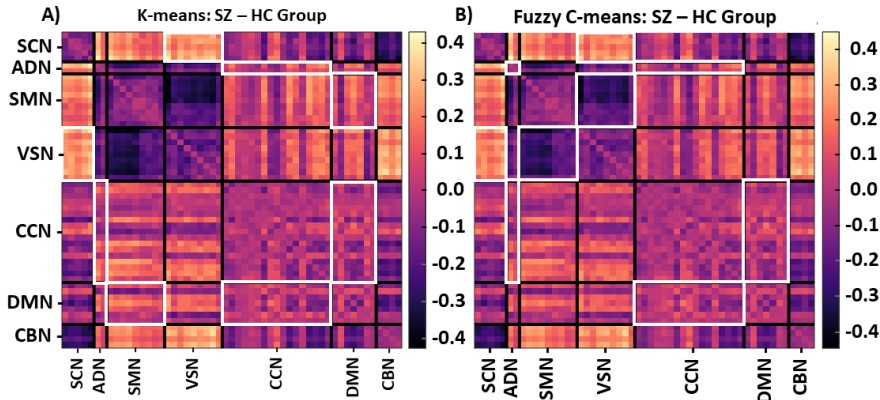

Figure 6: FNC Clustering Results. Panel A shows the result of the mean FNC of the SZ dominant k-means cluster minus the HC dominant cluster, and Panel B shows the result of the mean FNC of the SZ dominant fuzzy c-means cluster minus the HC dominant cluster. The colormaps to the right of each matrix show the magnitude of its values. The black grid denotes the boundaries of FNC domains, and the white rectangles indicate the domains that were identified as most important for the clustering.

to implement, and (2) they are widely applicable across clustering methods. These capabilities could enable our approaches to be highly impactful across the variety of fields in which clustering algorithms are used and for individuals with a wide range of data science expertise.

We apply both G2PC and L2PC to clusters generated by five popular clustering algorithms from low-dimensional synthetic data and demonstrate that they can work with each clustering algorithm to identify the expected importance of each features. In some instances with GMMs, several of the features that are expected to be important to the clustering of the synthetic data seem to be unimportant after analysis with G2PC and L2PC. G2PC and L2PC enable the comparison of feature importance across different clustering algorithms via a single percent change metric. As such, they could eventually be used to provide greater insight into the similarities and differences of clustering algorithms. This percent change metric is a vital and novel component of our adaptation. While G2PC and L2PC work well for low-dimensional data across all of the clustering methods with which we pair them, they were not as generalizable to high dimensional data as we first hoped.

To examine the utility of G2PC and L2PC for high dimensional data and for neuroimaging analysis, we apply all 5 clustering methods with G2PC and L2PC to FNC data extracted from FBIRN dataset and identify key features differentiating the SZ dominant cluster and the HC dominant cluster. However, very few of the individual features seem to have an effect upon the clustering. This is not entirely surprising given the high dimensionality of the data and that the Euclidean distance is used to cluster the data. This could indicate a limitation of G2PC and L2PC for high dimensional data. To control for this potential limitation, we implement a grouping component that allows related features to be perturbed or permuted simultaneously, and we obtain feature importance results for both k-means and fuzzy c-means clustering that are highly consistent. These consistent results support the reliability of the explainability methods for those clustering algorithms in high dimensions. The results obtained for G2PC and L2PC are also highly comparable to those for LR-ENR.

The grouping component of G2PC and L2PC is ideal for high dimensional datasets with features that are divisible into groups based upon domain knowledge. A known weakness of perturbation and permutation methods is that they can generate samples outside of the typical data distribution and produce unreliable results when a high degree of correlation exists between some features. The grouping parameter can help address this problem by grouping highly correlated features during analysis. When groups cannot be easily identified based on domain knowledge, an alternative approach would be to examine the correlation between each feature to find groups of features that are highly correlated or to randomly generate groups of features of size n.

Our results for the FBIRN FNC analysis agree with and extend existing literature. Four cross-network domain sets - ADN/CCN, SCN/VSN, CCN/DMN, SMN/DMN – and 1 network – ADN - are most important. This is consistent with Liang et al. (2006) who showed that schizophrenia is linked to widespread dysconnectivity across the brain. Additionally, our results show a disrupted (i.e., both increased and decreased) pattern in different brain networks. For example, we find VSN/SMN FNC is lower in SZ subjects than in HC ones. In fact, Chen et al. (2014) report a decrease in SZ subjects' SMN/VSN FNC relative to HC subjects. This could potentially explain the impairment of sensory information processing in SZ subjects. Multisensory information processing is a prerequisite for self-awareness. It has been proven that matching visual perception and proprioceptive signals from SMN are necessary for self-consciousness perseverance, as laid out in Ehrsson (2007). However, Medalia & Lim (2004) find that these signals are impaired in SZ subjects. Therefore, the disconnection among SZ sensory networks could potentially explain the underlying mechanisms of self-awareness deficit in SZ subjects.

We also find a decrease in ADN connectivity in SZ subjects, which aligns with findings in Li et al. (2019). This piece of evidence might explain the link that was found by Linszen et al. (2016) between hearing loss at an early age and the later development of schizophrenia. Interestingly, we find an increase in the ADN/CCN FNC in SZ subjects. Increased ADN/CCN FNC could serve as a compensatory mechanism and suggests a prospective study. We also found an increase in the FNC between VSN/SCN. This is supported by Yamamoto et al. (2018), who show an increased FNC between the thalamus (i.e., part of the SCN) and occipital cortices/postcentral gyri (i.e, part of the VSN) in SZ subjects compared with that of HC subjects.

Throughout each of our experiments, we utilize similar G2PC and L2PC settings. Different parameter settings would likely produce slightly different results, and it might be helpful for future studies to examine the effects of the parameter settings. We examined the effects of different values of K upon G2PC and L2PC in Appendix B. For L2PC, a decreased M would likely result in higher variance across repeats. However, the need for more reliable L2PC results must be balanced with its computational intensity. The ideal M is also likely affected by the characteristics of the dataset and clustering algorithm. Additionally, in our clustering analyses, we use Euclidean distance, which can be suboptimal for high dimensional data. This might explain why G2PC and L2PC did not have optimal performance in high dimensional data without grouping. Future studies could investigate the effects of other distance metrics and how they might enable G2PC and L2PC to be applied in higher dimensions.

It is likely that other model-agnostic explainability methods from the domain of supervised machine learning explainability could also be generalized to enable explainability for clustering methods. Permutation feature importance differentiates itself from many other model-agnostic explainability methods in that it can be applied to classifiers that output a hard classification. Additionally, we demonstrate a novel adaptation of perturbation that enables it to be extended to explain hard classification methods. In contrast, many model-agnostic explainability methods require that a classifier predict a class probability, rather than a binary class label. Given this requirement, soft-clustering methods like GMMs and fuzzy c-means could be ideal for compatibility with the majority of model-agnostic explainability methods. It is also feasible that G2PC could be used as a feature selection method to obtain optimal clustering similar to how Gómez-Ramírez et al. (2020) have used permutation for feature selection with supervised classifiers. Additionally, L2PC has the potential to be applied to explain supervised machine learning models like support vector machines (SVMs) by Boser et al. (1992).

## 5   Conclusion

In this study, we proposed - for the first time - the adaptation of model-agnostic explainability methods from the domain of supervised machine learning explainability to provide algorithm-agnostic clustering explainability. We introduce two new explainability methods that are both easily implemented and widely applicable across clustering algorithms. These capabilities could enable them to be impactful across many application areas and contexts. We demonstrate on low-dimensional, ground-truth synthetic data that they can be paired with multiple clustering algorithms to identify the features most important to differentiating clusters. We further demonstrate the utility of the methods for high-dimensional datasets by analyzing

rs-fMRI FNC data and identifying cross-domain connectivity associated with schizophrenia. It is our hope that this paper will (1) stimulate rapid growth in the domain of clustering explainability via an infusion of algorithm-agnostic methods from the domain of supervised machine learning explainability and (2) enable data scientists across a variety of fields to gain more insight into the variety of clustering algorithms that they employ.

### Acknowledgments

We thank those who collected the fBIRN dataset.

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

## A  Description of fBIRN Dataset

The fBIRN dataset contains 151 schizophrenia (SZ) subjects and 160 healthy controls (HC) that were collected from seven sites, including the University of California, Irvine; the University of California, Los Angeles; the University of California, San Francisco; Duke University/the University of North Carolina at Chapel Hill; the University of New Mexico; the University of Iowa; and the University of Minnesota collected neuroimaging data. Six 3T Siemens and one 3T General Electric scanner with the same protocol were used to collect the imaging data. T2\*-weighted functional images were collected using AC-PC aligned echo-planar imaging sequence with TE=30ms, TR=2s, flip angle = 77°, slice gap = 1 mm, voxel size= $3.4 \times 3.4 \times 4$ $mm^3$, and 162 frames, and 5:24 min. All participants were instructed to close their eyes during the rs-fMRI data collection. Neuroimaging data were preprocessed using statistical parametric mapping (SPM12[4]) in the MATLAB 2019 environment. We used rigid body motion correction to account for subject head movement. Next, the imaging data underwent spatial normalization to an echo-planar imaging (EPI) template in standard Montreal Neurological Institute (MNI) space and was resampled to 3x3x3 $mm^3$. Finally, we used a Gaussian kernel to smooth the fMRI images using a full width at half maximum (FWHM) of 6mm. The Neuromark automatic independent component analysis pipeline within the group ICA of fMRI toolbox GIFT[5] was used to extract 53 independent components (ICs), as described in Du et al. (2020).

## B  Analysis of Stability of G2PC and L2PC Importance across Repeats

To gain insight into how the number of repeats affected G2PC and L2PC importance across repeats, we repeated G2PC and L2PC to the previously generated synthetic datasets. Note that we only generated 1 set of data for both G2PC and L2PC, rather than the 100 sets of data that we generated for our first G2PC synthetic data experiments. The sets of synthetic data were identical to those used in our earlier L2PC synthetic data experiments. We ran G2PC and L2PC for 1000 repeats. For G2PC, we calculated the mean importance of each feature for each iteration and all previous iterations. For L2PC, we calculated

---

[4] `https://www.fil.ion.ucl.ac.uk/spm/`
[5] `http://trendscenter.org/software/gift`

the mean importance of each for each iteration and all previous iterations. The results are shown in Figure 7. In general, the importance values stabilized fairly quickly, though the specific clustering algorithm and dataset tended to affect the number of repeats necessary to obtain stable importance values. Depending upon the clustering algorithm, G2PC results stabilized within 100 to 400 repeats, and L2PC results generally stabilized within 200 repeats.

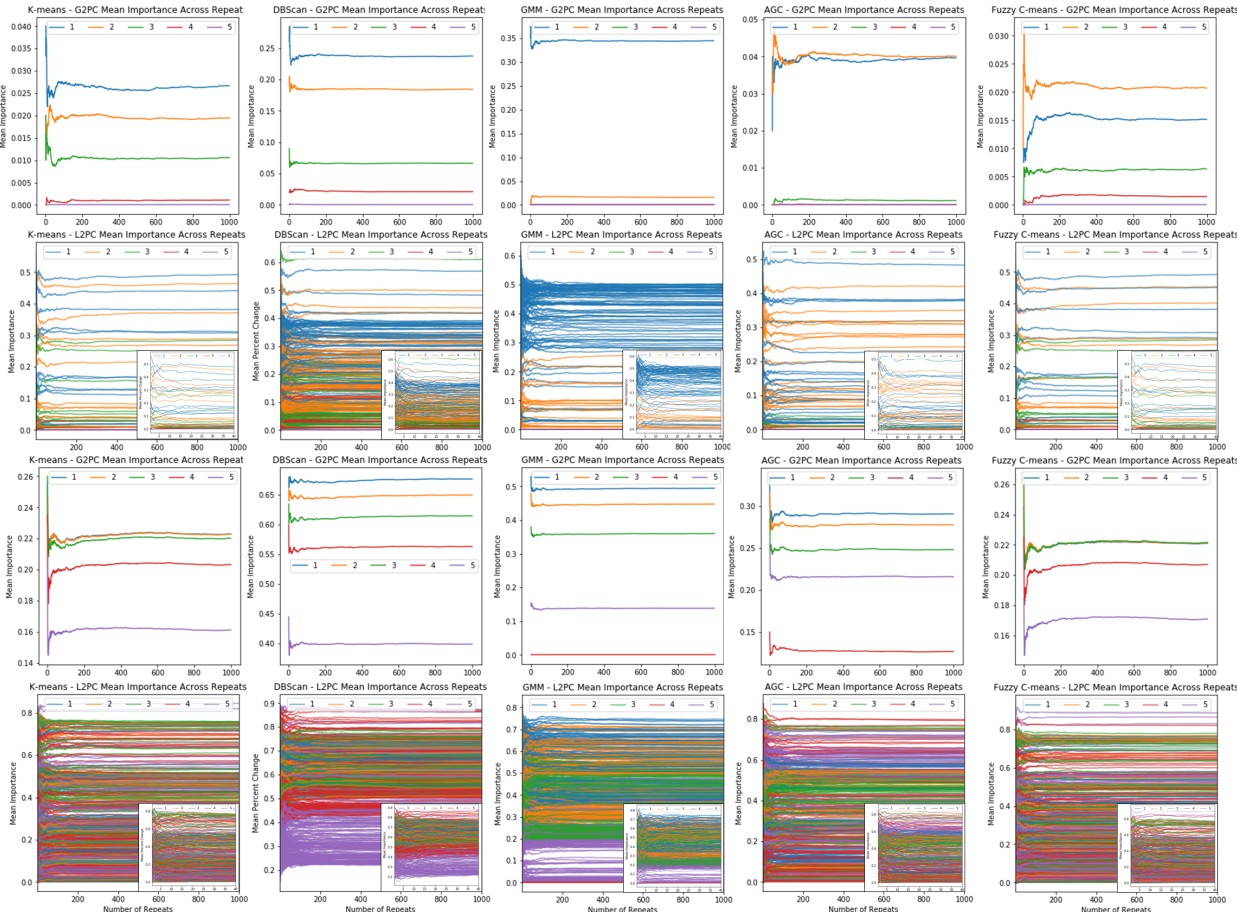

Figure 7: G2PC and L2PC Importance Across Repeats for Synthetic Data. The top two rows show the G2PC and L2PC results for Synthetic Dataset 1, and the bottom two rows show the G2PC and L2PC results for Synthetic Dataset 2. Each column shows the results for a different clustering algorithm. Because the L2PC importance values stabilized somewhat faster, each L2PC panel has a smaller inset panel that shows the mean importance for the first 40 repeats to provide further insight to make the variation across the first repeats.

