# OpenReview forum: "Algorithm-Agnostic Explainability for Unsupervised Clustering"
_TMLR — Rejected by TMLR_

### Review · Reviewer_ZX1Y · 2022-04-22

**Summary Of Contributions:**

The paper adapts the model-agnostic explainability methods to provide clustering explainability. For adaptation, the paper introduced two new explanability methods and presented the results on low-dimensional synthetic datasets and high-dimensional real world datasets. In the former, the presented explainability methods identified the most important features for differentiating clusters. While in the latter, they also showed inter-features connectivity.

**Requested Changes:**

The ideas presented in this work is interesting and as authors have pointed out could be impactful as well. However, I feel the algorithms, and the presented arguments are not verified rigorously. Here are the few questions/comments/suggestions:

1. Authors point out in multiple instances that their method is "easy to implement". Is this referring to small changes made to existing methods to propose new method or are authors referring to the whole method as "easy to implement". In any case, this warrants more explanation as this is presented as the important property of the proposed method.
2. The proposed method would require more verification. For instance, the future work directions suggested in the paper could acutally be done in this paper itself to demonstrate that the method is robust to these parameters? E.g., varying K, M and also use of different distance metric, etc.
3. Can authors show some empirircal analysis on the result presented on last paragraph of section 3.2? i.e., samples lying close to the boundary, sample frequency vs. pertubration sensititvity, etc.
4. Beside identifying the patterns learned by the clustering algorithms, can authors provide experiments on the ability of important features to perform clustering again?
5. What's the accuracy of clustering algorithms? Instead of speculating (3.1) if GMM and AGC have identified spuriuos clusters, can authors do some rigorous analysis to find out what's happening?
6. How would grouping parameter help to address unrealiable results caused by perturbation and permutation based methods generating out of distribution samples?
7. Missing comparision: as authors acknowledge that it is likely other methods could be generalized to enable explanaibility for clustering methods. Without any comparision, it is difficult to justify if the results' from presented method is superior.
8. The arguments made in the paper seems hand-wavy and would require more rigorous analysis. Please carefully proofread your paper and try to avoid the claims which can't be backed or require more experiments to support them which could be too much work for 1 journal. E.g.:
a. the grouping parameter can help address this problem by grouping highly correlated features,
b. can likely identify the patterns learned by the clustering algorithms,
c. may identify spurious clusters that are unrelated to the underlying groups, etc.
9. Can authors do experiment on other datasets as well? The result on real world dataset is interesting but for ML journal, instead of focusing too much on the domain, it would be more interesting to actually understand the behavior of the proposed methods.
10. Presentation: The overall presentation of the paper is only satisfactory and can be significantly improved. This applies for both text and presentation of results in tables and figures. E.g.,
a. What does color signify in Fig 4?
b. Can authors put the cluster results (3.2) in separate table?

Minor comments:
- In 2.1.3, L2PC "perturbs" should be L2PC "swap"?

**Strengths And Weaknesses:**

Strengths
- new application of explainability for clustering
- interesting application

Weakness:
- limited novelty of the algorithm (only few changes and that is not evaluated properly). Authors need to explicitly point out their contributions.
- limited clarity in the presented work (figures, description, etc.)
- lack of rigor in the arguments (e.g., "the grouping parameter can help address this problem by grouping highly correlated features", etc)

---

> ### Author Response · Authors · 2022-05-21
> **Response to Comments by Reviewer ZX1Y, Part 1**
>
> 1.	Authors point out in multiple instances that their method is "easy to implement". Is this referring to small changes made to existing methods to propose new method or are authors referring to the whole method as "easy to implement". In any case, this warrants more explanation as this is presented as the important property of the proposed method.
>
> We thank the reviewer for their question. The method as a whole is easy to implement. It only takes a few lines of code, and any mathematics involved are straightforward, easily comprehensible to researchers without a computer science background.
>
> 2.	The proposed method would require more verification. For instance, the future work directions suggested in the paper could actually be done in this paper itself to demonstrate that the method is robust to these parameters? E.g., varying K, M and also use of different distance metric, etc.
>
> It is challenging to generate synthetic data with many clusters and a predefined level of importance that is attributable to each cluster. However, we did assess synthetic data with 2 and 4 underlying clusters.
>
> The robustness to M or to the number of repeats is somewhat dependent upon the underlying data and clustering algorithm. Based on reviewer feedback, we added an additional analysis (Appendix B) examining the effect of increasing M.
>
> The method could be used with different numbers of k or different distance metrics. However, the method will potentially assign different levels of importance to different features if different distance metrics are used.
>
> 3.	Can authors show some empirircal analysis on the result presented on last paragraph of section 3.2? i.e., samples lying close to the boundary, sample frequency vs. pertubration sensititvity, etc.
>
> 4.	Beside identifying the patterns learned by the clustering algorithms, can authors provide experiments on the ability of important features to perform clustering again?
>
> We thank the reviewer for their suggestion. This would definitely be an interesting experiment to attempt. However, it might be a bit too much to add to this paper.
>
> 5.	What's the accuracy of clustering algorithms? Instead of speculating (3.1) if GMM and AGC have identified spuriuos clusters, can authors do some rigorous analysis to find out what's happening?
>
> We thank the reviewer for their comment. We updated Figure 4 to show the accuracies and updated the section 3.1 to discuss how the accuracy of the clustering might affect this.
>
> 6.	How would grouping parameter help to address unrealiable results caused by perturbation and permutation based methods generating out of distribution samples?
>
> We thank the reviewer for their comment. Perturbation and permutation methods are unreliable when highly correlated features are perturbed separately, resulting in perturbed samples that are highly irregular. By grouping and simultaneously perturbing features that are known to be related or correlated such that the grouped features maintain their internal relationships following perturbation, we can reduce the likelihood that out-of-distribution samples will be generated. I added some further explanation on this to the discussion.
>
> 7.	Missing comparision: as authors acknowledge that it is likely other methods could be generalized to enable explanaibility for clustering methods. Without any comparision, it is difficult to justify if the results' from presented method is superior.
>
> We thank the reviewer for their comment. The goal of this paper was to propose that model agnostic explainability methods could be generalized to the domain of clustering explainability. We demonstrated that two methods could be generalized and compared those methods. The generalization of other methods offers a future research direction but is beyond the scope of this work.
>
> 8.	The arguments made in the paper seems hand-wavy and would require more rigorous analysis. Please carefully proofread your paper and try to avoid the claims which can't be backed or require more experiments to support them which could be too much work for 1 journal. E.g.: a. the grouping parameter can help address this problem by grouping highly correlated features, b. can likely identify the patterns learned by the clustering algorithms, c. may identify spurious clusters that are unrelated to the underlying groups, etc.
>
> We thank the reviewer for their comment.
> G2PC and L2PC generally assigned the expected relative level of importance to features in our synthetic data experiments, which indicates that our method can identify patterns learned by the clustering algorithms (or at least identify the importance of each feature to the clustering algorithms).
> Thank you for pointing this out. We have made some updates to the paper based on your suggestions. (1) We calculated the accuracy associated with the clustering of the synthetic data to determine whether spurious clusters were identified. Please see the updated Figure 4 and Section 3.2.

---

> > ### Author Response · Authors · 2022-05-21
> > **Response to Comments by Reviewer ZX1Y, Part 2**
> >
> > 9.	Can authors do experiment on other datasets as well? The result on real world dataset is interesting but for ML journal, instead of focusing too much on the domain, it would be more interesting to actually understand the behavior of the proposed methods.
> >
> > 10.	Presentation: The overall presentation of the paper is only satisfactory and can be significantly improved. This applies for both text and presentation of results in tables and figures. E.g., a. What does color signify in Fig 4? b. Can authors put the cluster results (3.2) in separate table?
> >
> > We thank the reviewer for their comments. We updated the caption of Fig 4 to indicate the meaning of the different colors in the L2PC plots.
> > To clarify, is the reviewer suggesting that we convert Figure 5 to a table?
> >
> > Minor comments:
> > •	In 2.1.3, L2PC "perturbs" should be L2PC "swap"?
> > We thank the reviewer for their comment. We updated the text of 2.1.3 based on this comment with the hope that the updates would clarify the description of L2PC.

---

### Review · Reviewer_5cBZ · 2022-04-24

**Summary Of Contributions:**

This paper proposes two methods to explain unsupervised clustering. These methods compute feature importance based on permuting the features either globally with global permutation percent change feature importance (G2PC) or locally with local perturbation percent change feature importance (L2PC). While many methods for interpreting clustering results use decision trees to classify the cluster IDs, G2PC and L2PC these methods require training an additional model where there is often a trade-off between accuracy and explainability. G2PC and L2PC skip this additional step while still being to some extent “algorithm agnostic”.

These methods rank feature importance based on how often permuting a feature affects the cluster a point is assigned to. A key element to these two methods is it must be possible to classify new points to one of the existing clusters. This work proposes relatively natural hard classifiers for 5 popular algorithms: k-means, DBScan, a Gaussian mixture model, agglomerative clustering, and fuzzy c-means.

G2PC and L2PC are then evaluated on two synthetic datasets and a FMRI dataset. On the synthetic datasets, G2PC and L2PC are shown to recover the ground truth feature importance order across clustering algorithms for the most part. On the FMRI dataset, the feature importance values derived from G2PC and L2PC are shown to be overall consistent with those based on mean effect using linear regression.


**Requested Changes:**


According to the TMLR evaluation criteria, acceptance decisions should be based on the two questions:

Are the claims made in the submission supported by accurate, convincing and clear evidence?
Would at least some individuals in TMLR’s audience be interested in knowing the findings of this paper?

In order to satisfy the first criteria, I believe further justification of the claim that this is the first adaptation of model-agnostic explainability methods to clustering explainability is needed in two areas. First, that model agnostic explainability such as training a linear regression or tree-based classifier on top of the clusters of interest then performing standard supervised explainability procedures does not count as an adaptation to clustering. Second, that the proposed method is actually model agnostic and can be applied to all clustering methods including graph-based methods, or that the proposed classifiers are superior in some way such as that they provide better feature importance values or are more natural or accurate in some way.

In order to satisfy the second criteria, further experimentation on the benefits of permutation-based feature importance for clustering explainability is needed. Showing the behavior of G2PC and L2PC vs. Linear and Tree-based classifiers on a wider variety of datasets including generalizable analysis on the differences. It would also be interesting although in my view not necessary for acceptance, to explain the benefits of G2PC and L2PC over something that compares the means of clusters directly for explainability. Something like ranking feature importance based on the variance of cluster means seems like a straightforward method which has some drawbacks, but might be a simple alternative.


**Strengths And Weaknesses:**

Strengths:
* Clustering explainability is a relatively unexplored area with many of the current methods focusing on training interpretable decision trees as a supervised step after clustering. The proposed method skips this step by using some relatively natural models for classifying new points into the existing clusters which are applicable to many clustering algorithms.
* The approach is relatively simple conceptually, computationally inexpensive, and easy to implement making it an attractive method as compared to more complicated cluster-then-classify type approaches.
* The group permutation generalization seems promising in extending permutation-based approaches to higher dimensional settings where permutation-based feature importance methods are generally difficult to apply.
* The writing is for the most part clear with the exception of the algorithm specifications.

Weaknesses:
* One of the central claims of this paper is that G2PC and L2PC are “algorithm-agnostic” unsupervised clustering explainability methods. I do not believe this is true as both of these methods require the assignment of new samples to existing clusters as stated in section 2.1.4. In that section, it is shown how to build a model specific cluster classification model for 5 clustering algorithms, which breaks the algorithm-agnostic claim. While K-means, GMMs, and fuzzy c-means clustering all are defined by a small number of centers and can easily classify new points between these centers using a 1-nearest-neighbor classifier, for most other clustering models the choice of classifier is not so obvious. For example, for the family of graph-based clustering such as spectral clustering, Louvain clustering, Leiden clustering, it is unclear how to define a classifier for new points. For other models it is unclear if there exists a single preferred classifier, or if there are many reasonable ones. For instance, why should agglomerative clustering use a 1-nearest neighbor classifier over all points where K-means uses a 1-nn classifier to the means. Could we use 1-nn to cluster centers as a classifier for agglomerative clustering? Or why is this the “wrong” choice. Surely these would give different feature importance results in some cases. If so, we are baking in additional assumptions based on the extension of a clustering algorithm to new points, and these choices should be carefully considered.
* One of the claimed “key novelties” of G2PC (par 1, Page 4) is using the percentage of samples that switch clusters following the permutation of a particular feature instead of the ratio of the change in performance before and after permutation. I don’t see how this can be claimed as particularly novel. This is just using clustering accuracy as the performance function in algorithm 1.
* The experiments are limited such that it is difficult to draw generalizable insights about the proposed method. Only datasets with a small number of clusters (two four and two resp.) are tested leaving the question of the application to higher numbers of clusters in question. The grouping is tested only on the FMRI data, and seems to be a way of lowering the effective dimension of features explored, but is not explored beyond generally and seems to be specifically applicable to FMRI data with accepted ground truth mapping and feature clusters. The method is compared to linear regression “mean effect” trained on the clustering labels. This is an extremely useful comparison, but deserves more exploration. In my mind this is a valid alternative method of creating a classifier from a clustering and the benefits / drawbacks of G2PC and L2PC vs. this method would be interesting for someone looking to explain a clustering.
* Code is not available for review, although there appears to be a link to a GitHub repo in the paper, it is not a valid link.

Other comments / Questions:
Figures the L2PC results in figures 4 and 5 exhibit remarkably different structure. I was initially confused why line plots were used as it seems to suggest some sort of continuous relationship between the features, but when I realized that each line shows the L2PC of a datapoint, I was intrigued how in the synthetic data there are some points that are close to the boundary and permuting any feature will flip its class and some points that almost no feature will flip the class, whereas in the FMRI data there seem to be no points that are universally close to the boundary in all dimensions, qualitatively the lines in 5C, 5D are “jagged”. This seems to imply that there is something interesting going on here. Are there any generalizable insights that can be drawn from the data or model from these differences in the L2PC plots? (Note it would be great to add a note in the figure caption that each line represents a datapoint.)

Do all of the clustering methods achieve perfect accuracy to the ground truth clusters in the synthetic datasets?

Why are only k-means and c-means clustering shown on the FMRI data?

I assume the logistic regression “Mean effect” is the mean absolute value of the coefficients for plots Figure 5 E, F. In any case this should be clarified.

Minor points:

Algorithm formatting can be improved some notes below:
* Algorithm 1 contains several inconsistencies and undefined quantities
* Add back the vowels in mdl for clarity if you're going to keep "performance" in algorithm 1
* Y_0 is not defined, performance(Y_0, Y_1) is not, I assume Y_0 is the ground truth so should be Y
* J,K are inputs to the algorithm?
* Algorithm 3 is titled "Local Permutation ..." should be local perturbation?
* Algorithm 3 permute group is named the same as in algorithm 2 but is different. Perhaps local permute group?
* X_2 means different things in Algorithm 2 vs. 3.
* The notation on Algorithm 3 line 3 is unclear to me p < M – 1 means what exactly?
* The comment on line 5 of procedure L2PC says M x F, isn’t this 1 x F?
* It would be helpful to include what all these inputs are, while the “j in J groups” etc. is helpful, could a comment be added with a short description of all parameters? E.g. Algorithm 2 gives a sense (without looking at the text of J, N, and K, but not M, C, mdl, or groups.

Inconsistent spacing in the appendix.

Inconsistent usage of hyphens instead of dashes throughout.

FMRI dataset is not public and requires IRB approval thus increasing the bar for reproducibility. An experiment on a publicly accessible dataset would help the reproducibility of this study.

---

> ### Author Response · Authors · 2022-05-21
> **Response to Comments by Reviewer 5cBZ**
>
> •	One of the central claims of this paper is that G2PC and L2PC are “algorithm-agnostic” unsupervised clustering explainability methods...
>
> We thank the reviewer for their comment. This is definitely an interesting point and bears further consideration. While it is true that G2PC and L2PC require that perturbed samples be somehow reassigned to existing clusters, that does not necessarily mean that the methods are not algorithm-agnostic or that they are not broadly applicable to different clustering methods. Within explainability for supervised classifiers, methods are commonly considered model-agnostic even though they require using the specific classifier for reclassifying perturbed samples. It is possible that assigning samples to existing clusters by completely redoing the clustering and using a slightly different metric that examines the percent of samples that are no longer clustered together following perturbation could be considered more algorithm-agnostic. For agglomerative clustering, using a 1-nn classifier to the means could work. However, doing so would depart more from how agglomerative clustering works by looking for the nearest sample or group of samples than assigning a perturbed sample to the cluster of the nearest unperturbed sample.
>
> •	One of the claimed “key novelties” of G2PC (par 1, Page 4) is using the percentage of samples that switch clusters following the permutation of a particular feature...
> The biggest contribution of this work is making the case that model agnostic explainability methods for supervised classification can be easily generalized to explain clustering algorithms and thus could rapidly accelerate the domain of explainability for clustering algorithms. The percent of samples that change clustering is novel and is one of the key adaptations that makes it possible to apply permutation feature importance and perturbation to clustering algorithms.
>
> •	The experiments are limited such that it is difficult to draw generalizable insights about the proposed method. .
> There are benefits and drawbacks to both methods. Using logistic regression requires that a new model be trained (ideally for multiple folds), which can be computationally intensive when there are a large number of dimensions. In contrast, G2PC and L2PC generally do not require any additional training. Samples can just be assigned to existing clusters. When performing analyses with high dimensions, it is possible that some dimensions might be ignored by logistic regression because the use of other features is sufficient to perform accurate classification. In this case, logistic regression might not give the best explanation. However, it does give a level of importance for each individual feature. With G2PC and L2PC at high dimensions, the perturbation of an individual feature is unlikely to have a strong effect, so it is necessary to group features.

---

> > ### Author Response · Authors · 2022-05-21
> > **Response to Comments by Reviewer 5cBZ  Part 2**
> >
> > •	Code is not available for review, although there appears to be a link to a GitHub repo in the paper, it is not a valid link.
> >
> > We thank the reviewer for pointing this out. We have now updated the link to the GitHub repo.
> >
> > Other comments / Questions: Figures the L2PC results in figures 4 and 5 exhibit remarkably different structure...
> >
> > We thank the reviewer for their comment. I updated the L2PC figure captions to indicate that each line is for an individual sample.
> > It is possible that generalizable insights might be drawn from differences in L2PC plots. For example, if a larger number of samples belonging to one cluster switch clusters following perturbation, then it is possible that that cluster is more diffuse than the other clusters or that it has more samples closer to the cluster boundary.
> >
> > Do all of the clustering methods achieve perfect accuracy to the ground truth clusters in the synthetic datasets?
> >
> > We thank the reviewer for their comment. We updated Figure 4 to show the accuracies and updated the section 3.1 to discuss how the accuracy of the clustering might affect this.
> >
> > Why are only k-means and c-means clustering shown on the FMRI data?
> >
> > We thank the reviewer for their question. We partially address this at the start of Section 3.2. Perturbing groups of features in high dimensions did not have a significant effect for GMMs and Agglomerative Clustering. DBScan generally wasn’t able to identify clusters in the fMRI dFNC data, so we couldn’t apply G2PC and L2PC to it.
> >
> > I assume the logistic regression “Mean effect” is the mean absolute value of the coefficients for plots Figure 5 E, F. In any case this should be clarified.
> >
> > We thank the reviewer for raising this point. This point is clarified in the second paragraph of Section 2.2.4. We updated the paragraph with a sentence to indicate that we took the absolute value of the coefficients multiplied by their respective samples.
> >
> > Minor points:
> > Algorithm formatting can be improved some notes below:
> > •	The notation on Algorithm 3 line 3 is unclear to me p < M – 1 means what exactly?
> > In the line immediately beforehand, we generated an array of permuted values between 0 and the total number of samples minus 1. We then used indexing to select samples with corresponding permuted indices below the total number of perturbations (M).
> > •	The comment on line 5 of procedure L2PC says M x F, isn’t this 1 x F?
> > No. X_2 has dimensions equivalent to the number of perturbations (M) by the number of features (F).
> >
> > Inconsistent spacing in the appendix.
> >
> > We thank the reviewer for their comment. We updated the spacing in the appendix.
> >
> > Requested Changes:
> > In order to satisfy the first criteria, I believe further justification of the claim that this is the first adaptation of model-agnostic explainability methods to clustering explainability is needed in two areas...
> >
> > We thank the reviewer for their comment. To clarify, training a linear model or tree-based classifier is not generally considered a model-agnostic explainability method (though this approach would be related to some model-agnostic methods like LIME). As we showed in the paper, it is definitely possible to train an interpretable classifier on top of the identified clusters. However, using perturbation or permutation approaches has some advantages. They operate directly on the underlying data and clustering algorithm. Training an interpretable model on top of the identified clusters risks that the model will find a way to assign the samples to their underlying clusters without actually relying upon all of the features that were used by the clustering algorithm. That being said, an interpretable classifier (e.g., logistic regression) could give insight into the importance of each feature in high dimensions. The importance may just not capture exactly what the clustering algorithm captured.
> >
> > We demonstrated that G2PC and L2PC are broadly applicable to 5 different categories of clustering algorithms. Each of these 5 clustering method categories operate directly upon the underlying samples or upon edges as we demonstrated in our between-graph (i.e., dFNC clustering), so perturbed samples and graphs can be easily reassigned to an existing cluster. G2PC and L2PC would not be applicable to within-graph graph-based clustering of nodes.
> >
> > Within the domain of supervised model explainability, many explainability methods are considered model-agnostic, even if they are not necessarily applicable to every classifier. For example, PD plots and ICE plots are considered model-agnostic explainability methods, but they are not applicable to classifiers like SVMs that do not output a probability of a sample belonging to a particular class. Similarly, G2PC and L2PC are algorithm-agnostic in the sense that they broadly applicable to any clustering algorithm that operates directly on the underlying samples. However, they are not applicable to within-graph graph-based clustering.

---

> > > ### Author Response · Authors · 2022-05-21
> > > **Response to Comments by Reviewer 5cBZ Part 3**
> > >
> > > In order to satisfy the second criteria...
> > >
> > > We thank the reviewer for their comments. While using additional datasets with different characteristics in the paper could definitely provide greater insight into the varying performance of G2PC/L2PC and interpretable classifiers, doing so would also greatly increase the length and complexity of the paper.

---

### Review · Reviewer_dk4q · 2022-04-27

**Summary Of Contributions:**

The authors propose a straightforward permutation based approach(es) to assessing feature importance for clustering. The approach is (almost, see below) model agnostic. They apply their approach to 5 clustering algorithms and simulated and fMRI data.

**Broader Impact Concerns:**

None noted.

**Requested Changes:**

The main concerns I would like addressing are:
1. Cluster labels could change so shouldn't the clusterings be compared based on pairwise membership? The authors recognize this in section 2.1, but their solutions for aligning cluster labels across repeats means that the approach is no longer truly model agnostic, which is given as a key selling point of the approach. In particular it is unclear how you do this for graph-based clustering methods or something like affinity propagation. I think a better metric would be something like mean((c1_i==c_1j)==(c2_i==c2_j) for all i,j) where i,j index all pairs of samples and c1 and c2 are cluster assignments for the original data and permuted data respectively.
2. Why not permute each feature in a group independently? What is the trade off here?
3. Would be good to include a graph-based clustering method since these are popular in single cell genomics.
4. "null hypothesis of zero perturbation percent change" what does this mean? How would generate this null? Cut this if you don't know.
5. In the fMRI data the authors note that permuting individual features did not affect clustering much, and so resorted to permuting (known) groups of features. What happens if individual features don't affect the clustering but you don't have known groups to permute? (I think sampling random groupings and averaging over these could help).

Not a concern/requirement for publication but I do think it would be interesting to compute and compare to saliency maps for the assignments. At least for soft k-means/GMM I think it would be reasonably straightforward (and computationally cheap) to get the gradient of the soft assignment wrt to the features.

Minor comments:
Algorithm 1. I'm not sure it's worth separating out the PERMUTE_FEATURE function, it could just be one line as
X2[:,j]=X2[random_permutation(1:N),j]
why do you need "features" rather than just indexes X directly?
It's unclear if X,Y are for the whole dataset or one data point (possibly either?) This becomes clear later but should be clear on first reading.

change "randomly distributed data" to "simulated datasets"

"This indicates that those samples may be closer to the boundaries of their clusters"... they may also just be noisier samples for some reason.

**Strengths And Weaknesses:**

The paper is well written although substantially longer than it needs to be, in particular the intro and discussion. The introduction is good but sometimes a little repetitive so could be edited to be more succinct. I also find the citation style a little strange, e.g. "logistic regression, which was introduced by Cox (1958)" rather than "logistic regression (Cox 1958)". Not a big deal but does make it wordy than need be. I don't really like discussions that just repeat the paper. It should be a discussion of additional thoughts/ideas (as the last paragraph does).

The technical contribution is not highly significant: permuting features is about the simplest approach you could think of for assessing importance and their is no theoretical justification given (as opposed to SHAP for example for supervised learning). However, it is a very reasonable approach and worth exploring in practice as the authors have done.

---

> ### Author Response · Authors · 2022-05-21
> **Response to Reviewer dk4q**
>
> 1.	Cluster labels could change so shouldn't the clusterings be compared based on pairwise membership? The authors recognize this in section 2.1, but their solutions for aligning cluster labels across repeats...
>
> We thank the reviewer for their comment. This is a good point. In the specific clustering algorithms that we employed, it was possible to assign samples to existing clusters, so it was not a concern that the underlying cluster labels would change (e.g., that cluster 0 would become cluster 1 or that cluster 2 would become cluster 4). For graph-based clustering methods, a metric similar to what you described might be feasible. Please see our thoughts related to graph-based clustering included in our response to the third concern that the reviewer mentioned for further discussion on this topic.
>
> 2.	Why not permute each feature in a group independently? What is the trade off here?
>
> We did try permuting each future in a group individually. However, when we only perturbed 1 out of 1378 features at a time, each feature had very little affect. However, when we perturbed groups of features, that had a significant effect. Within each group, we perturbed each feature in a particular group for a sample by replacing their values with the values from another sample. One weakness of perturbation methods is that they can produce out-of-distribution samples, especially when certain features are highly correlated. With our perturbation approach, we maintained the relationship between features in the same group (which are more likely to be highly correlated), reducing likelihood of generating out-of-distribution samples that adversely affected explanations.
>
> 3.	Would be good to include a graph-based clustering method since these are popular in single cell genomics.
>
> We thank the reviewer for their comment. Each of the 5 clustering method categories utilized in this paper operate directly upon the underlying samples (e.g., our synthetic data examples) or upon the edges of existing graphs (e.g., our dFNC data analysis), so perturbed samples or graphs can be easily reassigned to an existing cluster. As such, G2PC and L2PC can be applied with between-graph graph clustering methods in which the underlying graph edges could be treated as features and perturbed. However, they would not be applicable to within-graph graph-based clustering of nodes. Would the reviewer recommend that we add an analysis of this nature?
>
> 4.	"null hypothesis of zero perturbation percent change" what does this mean? How would generate this null? Cut this if you don't know.
>
> If a particular feature or feature group is unimportant, its perturbation could be expected to have no effect. As such, it would result in zero perturbation percent change. Other statistical approaches could potentially be applied instead, but this would be a useful first analysis to determine whether the effect of perturbing a feature or feature group is statistically significant.
>
> 5.	In the fMRI data the authors note that permuting individual features did not affect clustering much, and so resorted to permuting (known) groups of features. What happens if individual features don't affect the clustering but you don't have known groups to permute? (I think sampling random groupings and averaging over these could help).
>
> One could seek to identify groups of similar features. One could do this by calculating the correlation between individual features and identifying groups of features with high levels of correlation. We updated the paper discussion to address this.
> Not a concern/requirement for publication but I do think it would be interesting to compute and compare to saliency maps for the assignments. At least for soft k-means/GMM I think it would be reasonably straightforward (and computationally cheap) to get the gradient of the soft assignment wrt to the features.
>
> This is a really good idea and would be super interesting to do, as the probabilities associated with soft clustering methods creates a lot of opportunities to apply model agnostic methods from supervised classification. We have plans to do something similar to this for a separate study in the next few months.
>
> Minor comments: Algorithm 1. I'm not sure it's worth separating out the PERMUTE_FEATURE function, it could just be one line as X2[:,j]=X2[random_permutation(1:N),j] why do you need "features" rather than just indexes X directly? It's unclear if X,Y are for the whole dataset or one data point (possibly either?) This becomes clear later but should be clear on first reading.
> change "randomly distributed data" to "simulated datasets"
>
> We thank the reviewer for their comment. We updated the paper to reflect this recommendation.
>
> "This indicates that those samples may be closer to the boundaries of their clusters"... they may also just be noisier samples for some reason.
>
> We updated the paper to include this comment.

---

> > ### Comment · Reviewer_dk4q · 2022-05-21
> > **Graph clustering and stats**
> >
> > Re: graph clustering. I should have made this clearer. Where I see this used the most is in single cell genomics. Here the aim is to clusters cells (nodes of the graph) into cell types. The graph is a kNN constructed from per cell features (gene expression). So although the clustering is graph based, there are underlying features that could could permute in your method(s), which would change the kNN and thereby the clustering. The most commonly used implementation is probably https://satijalab.org/seurat/reference/findclusters
> >
> > Re: statistical testing (point 4). I still don't think you have proposed a way of asking whether whether the difference from 0 is statistically significant. e.g. you could (repeatedly) add a dummy feature(s?) that you know is/are random noise, and get the perturbation % changes as an empirical null for "unimportant" features. I'm not saying this is the right scheme, but the sort of thing you could think about doing.

---

> > > ### Author Response · Authors · 2022-05-30
> > > **Graph clustering and stats**
> > >
> > > Re. graph clustering. Thank you for clarifying. If we were to do what you are suggesting with graph clustering, would we need to cluster all of the data again after every perturbation, rather than assign perturbed samples to existing clusters like we are currently doing?
> > >
> > > Re: statistical testing. Thank you for your suggestion. That is an interesting idea that could definitely be a viable option. We could also potentially perturb multiple randomly selected groups of features and calculate their perturbation percent change. The randomly selected groups could have the same number of features as the real groups of interest. If the groups of interest had significantly more perturbation percent change than randomly selected groups, the groups of interest might be considered significantly important.

---

### Decision · Action_Editors · 2022-06-08

**Recommendation:** Reject

**Comment:**

The paper was reviewed by three reviewers, and the reviews were both comprehensive and well-aligned with the TMLR evaluation criteria. The authors engaged with the reviews and provided responses to all of the reviewer's comments.

The reviews pointed out several strengths that highlighted the importance of tackling the problem of feature relevance in unsupervised learning problems. At the same time, there were several concerns that the reviewers identified that, if addressed, would substantially improve the paper.

In particular, one reviewer suggested "their solutions for aligning cluster labels across repeats means that the approach is no longer truly model agnostic". The claims of a model-agnostic approach might be revisited to align with the methodology and evidence. Another review raised concerns about the alignment between the claims and the evidence supporting those claims. Finally statements such as "the grouping parameter can help address this problem by grouping highly correlated features" should be clarified and made more specific if not mathematically rigorous.


The suggestions provided by the reviews would greatly improve the manuscript and I would be glad to consider a substantially revised version of the manuscript.